# The Analysis of Trade Liberalization and Nutrition Intake for Improving Food Security across Districts in Indonesia

**Maya H. Montolalu** [1,2], **Mahjus Ekananda** [1,*], **Teguh Dartanto** [1], **Diah Widyawati** [1] and **Maddaremmeng Panennungi** [1]

1   The Department of Economics, Faculty of Economics and Business, University of Indonesia, Depok 16424, West Java, Indonesia; mayahmr2000@gmail.com (M.H.M.); teguh.dartanto@ui.ac.id (T.D.); diah.widyawati@gmail.com (D.W.); maddaremmengp@gmail.com (M.P.)
2   Socio-Economics Departement, Faculty of Agriculture, Sam Ratulangi University, Manado 95115, North Sulawesi, Indonesia
\*   Correspondence: mahyusekananda@gmail.com

**Abstract:** The debate on the effect of trade liberalization on food security poses solid arguments, both in favor as well as against the issue. This study aims to analyze the linkages between trade liberalization (measured using food import tariff exposure) and food security (measured using nutrition intake) in the case of Indonesia. The national food import tariff is decomposed into district-level import tariff exposure and is analyzed based on sectoral tariffs such as agriculture tariffs and food manufacture import tariffs. The analysis employs panel data of 496 Indonesian districts and postulates an association between trade and food security by using fixed-effect regression. By analyzing the effects of tariff exposure towards food consumption in all districts and grouping the districts into 5 (five) islands, we can contribute to the literature on trade liberalization and food security. First, it is found that import tariff exposure is negatively impacting nutrition intake and each sector has a different effect on each nutrition intake. Furthermore, the impact of manufacturing tariffs on calorie and protein intake is slightly higher than that of agriculture tariffs. Second, it is shown that both sectoral import tariffs' effects vary across islands in Indonesia. Furthermore, the research is expected to contribute to and become a reference for the government in regulating tariffs and other trade liberalization schemes to support households to be food secure.

**Keywords:** import tariff; tariff exposure; food security; nutrition intake

## 1. Introduction

Although global population growth and food growth are almost equal, the disproportion between regions with high food production and population distribution has caused an imbalance in access to nutrition. For instance, the share of people residing in industrial countries is less than that of those who live in lower-income countries, and populations living in the latter may be unable to share in the abundance of food because of their lack of purchasing power. Accordingly, it is projected that the world population will increase three times in 2050, which means more food demand will arise. It is stated that human numbers are increasing and food supply is also increasing; however, it is unclear whether food production can keep up with the pace of population growth [1].

In the era of globalization, providing and supplying adequate food and nutritious food to the household level is challenging for a country. The definition of food security according to the Food and Agriculture Organization (FAO) is "All people at all times have both physical and economic access to sufficient, safe and nutritious food to meet their dietary needs and food preference for their active and healthy life". It is a measure of the availability of food and individuals' accessibility to food, where accessibility includes affordability. Based on the FAO definition of food security, there are four dimensions of food

security such as food availability (on farms and markets), accessibility (by all households), utilization (function of food safety, nutritional status, and health), and food stability.

In developing countries, food insecurity and malnutrition have become a challenge in which a lot of people still suffer from this condition [2]. Thus, food security is not only a matter of individuals and households, but it is also important at the national level, where the government holds the responsibility of fulfilling the food needed at a reasonable price. It is also believed that trade can become an important element in achieving food security in which food trade and trade policy may encourage people, either producers or consumers, to utilize the available resource economically [3]. Therefore, trade can potentially bridge this mismatch and impact a country's development process, either directly or indirectly, at the macro and micro levels.

There are two basic reasons why countries engage in trade: first, countries benefit from their mutual differences, either geographically or with regard to their economic resources; and second, they want to achieve economic scale in production. Ricardian's model, which is the theory of comparative advantage, explains the gains arising from international trade, driven by the differences in each country's productivity and opportunity cost [4].

Trade liberalization enables countries to produce more goods and services because they have a comparative advantage. Moreover, trade improves countries' access to larger markets and, subsequently, improves their capacity to specialize in their production. There are two categories of countries that benefit from trade openness: first, countries where the poor constitute the majority net buyer of food and food imports are restricted; and second, countries where the poor constitute the net seller, while food exports are restricted [5].

The effect of trade on food security is still debatable. Many studies on trade have focused more on its relationship with growth, inequality, and poverty [6–8]; only several studies relate trade impacts to food security [9–11]. Yet, the World Health Organization (WHO) and the Food and Agriculture Organization (FAO) have emphasized the association between both poverty and food security with nutrition security [12], and trade has been highlighted as a critical macroeconomic driver that must be considered in improving diets, nutrition, and chronic disease prevention. In particular, trade liberalization has been found to increase economic activity and reduce poverty [13].

Nonetheless, the effect of trade on food security, to some extent, has drawn many researchers' attention [14–16]. For instance, research on the effect of import bans on households as consumers was done [17] in which import restrictions measured through tariffs resulted in price increases. The results showed that the poorest households will bear the burden of import reduction policies because they are net food buyers who typically spend a large proportion of their budget on basic food consumption [17]. On one hand, a prior study found that trade liberalization positively contributes to food security and the stability of food supply in Bangladesh [18].

With ever-increasing globalization, ensuring adequate availability and supply of food and nutrition at the household level is a major challenge for countries to overcome. In identifying the core reasons why food consumption is characterized by malnutrition [19], it is reported that households simply lack income sources to afford sufficient food. Thus, increases in food prices harm poor consumers due to the high proportion of their budget spent on food, and rising prices thereby result in rising poverty. Food security is closely related to consumption and consequently, to household poverty. Hence, any resources that have the potential to raise household income and benefit the poor would enhance food security and thereby, nutrition.

Therefore, trade not only improves household access to food by way of increasing their income but also enhances food availability by ensuring affordable prices. Reduced tariffs can boost the global food trade, subsequently increasing the quantity of food available. On the other hand, it is argued that it might also adversely affect domestic production because importing countries may be vulnerable to the volatility of global prices and supply [11]. The study is about the effect of trade on food security in China, which finds that international trade increases China's dependence on food imports and negatively affects

the country's food security. Another study [20]—an empirical examination of the effect of trade liberalization on food availability in 37 developing countries using dietary energy supply (DES) per capita as a measure of food security—also finds that trade liberalization exerts a negative influence on food security in the short run.

These mixed findings indicate that there still remains the empirical question of whether trade liberalization has a positive or negative effect on the dimension of food security and through what channels do those trade impacts travel to affect food security. In this study, the relationship between those two factors—tariffs and food security—will be explored by analyzing the effects of tariffs on food nutrition (calorie and protein) intake. We examine the effects at a regional level in Indonesia to obtain a richer picture of the dynamics in each district, as the variation in geographical and cultural characteristics at the sub-national level may contribute to different concentrations of effects in particular areas.

The rest of the paper is organized as follows. The next section covers the theoretical background, followed by Section 3, which presents the methodology used in this study. Section 4 presents the results and discussion, and the last section concludes the paper.

## 2. Theoretical Background

### 2.1. Overview of Indonesia's Trade Liberalization and Food Security

Nowadays, trade policy among countries has been governed by the General Agreement on Tariffs and Trade (GATT)—an international treaty—for almost 70 years. Since 1994, the World Trade Organization (WTO) has enforced international trade rules, through which it can tell the country if its policies have violated the agreements [4]. Nonetheless, each country has its own unique shape of policy intervention regarding trade and food security.

Globally, the number of people that suffer from being undernourished is increasing, which is revealed in the world prevalence of undernourishment (PoU) index. Specifically the PoU index of Indonesia is slightly increase from 6.4% in 2018 to 6.5% in 2019 (see Figure 1). This means that more people around the world are unlikely to have enough food that meets the dietary requirement. In Indonesia, the prevalence of people who suffer from insufficient food consumption has slightly increased from 8.32% to 8.34% from 2017 to 2020 (it is based on BPS data).

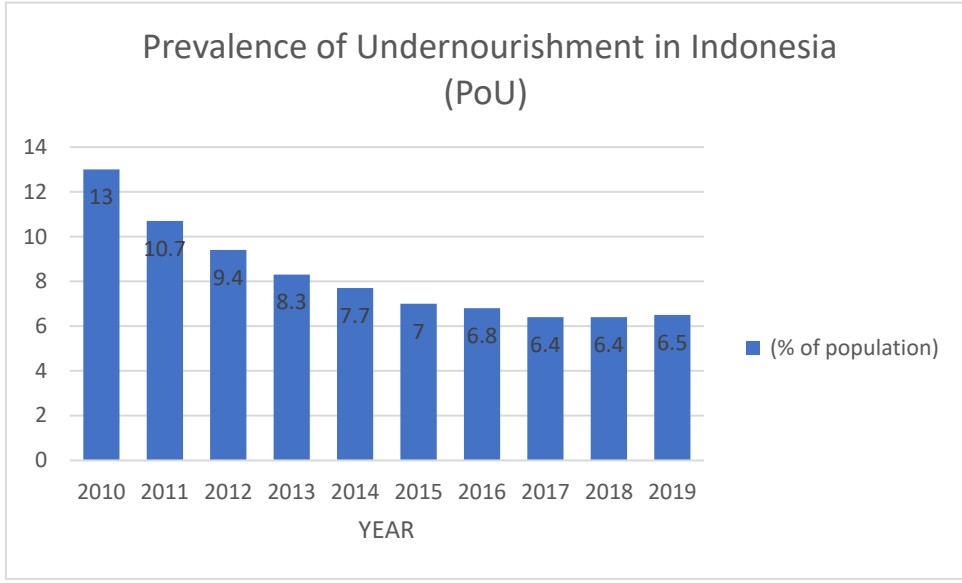

**Figure 1.** The PoU index of Indonesia from 2010-2019. Source: authors modified the chart from World Bank Data 2022. (https://data.worldbank.org/indicator/SN.ITK.DEFC.ZS?locations=ID, accessed on 6 March 2022).

Food and nutrition security policies in Indonesia are influenced by regional and international commitments and initiatives such as the Millennium Development Goals

(MDGs), the Association of South-East Asian Nations (ASEAN) Integrated Food Security Framework, the ASEAN Plus Three Emergency Rice Reserve, the Zero Hunger Challenge, and Scaling-Up Nutrition (SUN).

In addition, based on the global food security index (GFSI)-the index measures food security across most of the countries in the world- data shown in Figure 2, the index of GFSI Indonesia is fluctuating in an increasing trend. It represents the overall national food security index in which the rate is actually varied across regions based on the geographical circumstances, wealth and natural resources of the region.

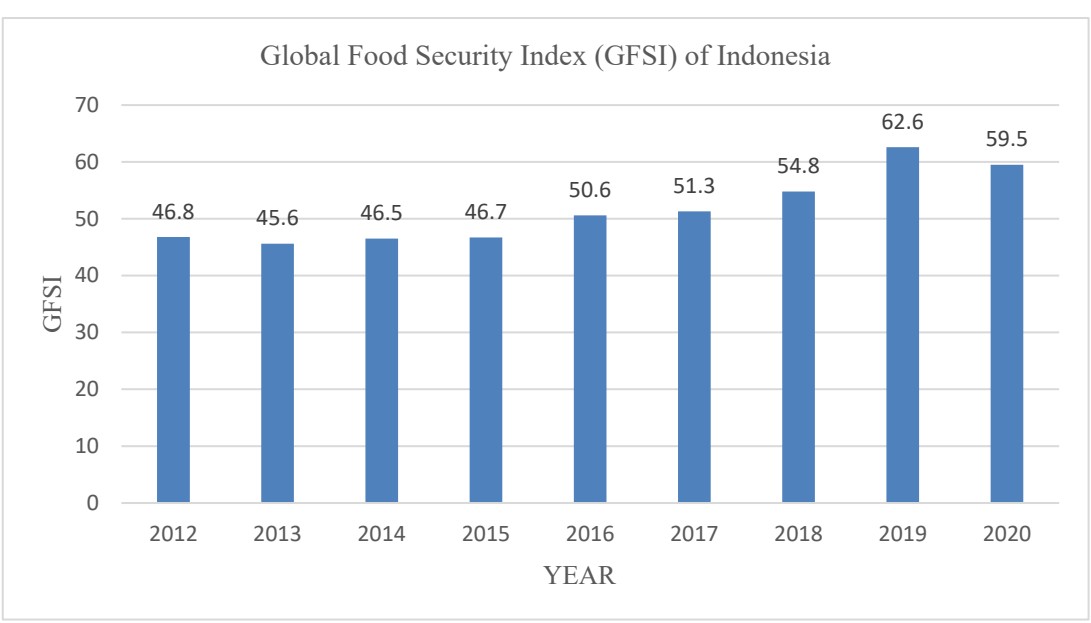

**Figure 2.** GFSI of Indonesia 2012–2020. Source: The Economist Intelligence Units, Databox. (https://databoks.katadata.co.id/datapublish/2021/02/26/ketahanan-pangan-indonesia-menurun-pada-2020, accessed on 12 February 2022).

Thus, in general, the food security index reveals an increasing trend; however, there are some regions where the overall food intake is still below the national threshold of food dietary needs. It is estimated that 38% of Indonesians cannot afford food that meets the dietary needs as suggested in the nutrient guideline. Based on the data of Statistics Indonesia, the poor are more likely to live in the eastern part of Indonesia; for instance, the highest poverty rate is found in Papua and West Papua which accounted for 26.64% and 21.37%, respectively [21]. Accordingly, Indonesia not only needs to increase consumption but also the variety of nutritious food intake.

Indonesia has participated in multilateral and regional trade agreements. The transformation of Indonesia's trade policy has fallen into five phases [22]. First, the era when trade was controlled through import bans, quotas, and tariffs, which ended when trade began to be normalized; second, the era when the oil boom induced the government to implement import substitution policies and move away from its dependence on oil exports; third, the era when the government adopted an aggressive export diversification strategy, which coincided with the establishment of the ASEAN Free Trade Area and the formation of the World Trade Organization (WTO); fourth, the era characterized by the domination of IMF program along with a removal of all import restrictions, the reductions of tariffs and importing of agricultural products, in addition to changes in major institutional mechanisms, the establishment of Bulog (The National Logistic Agency) and the initiation of the ASEAN Economic Community (AEC) and Free Trade Agreement (FTA); and fifth, the current era, in which trade policy has been simplified by reducing trade restrictions and improving transparency.

In addition to trade openness in Indonesia as a percentage of GDP, the trend can be seen in Figure 3, which shows a fluctuation from 2010 to 2020.

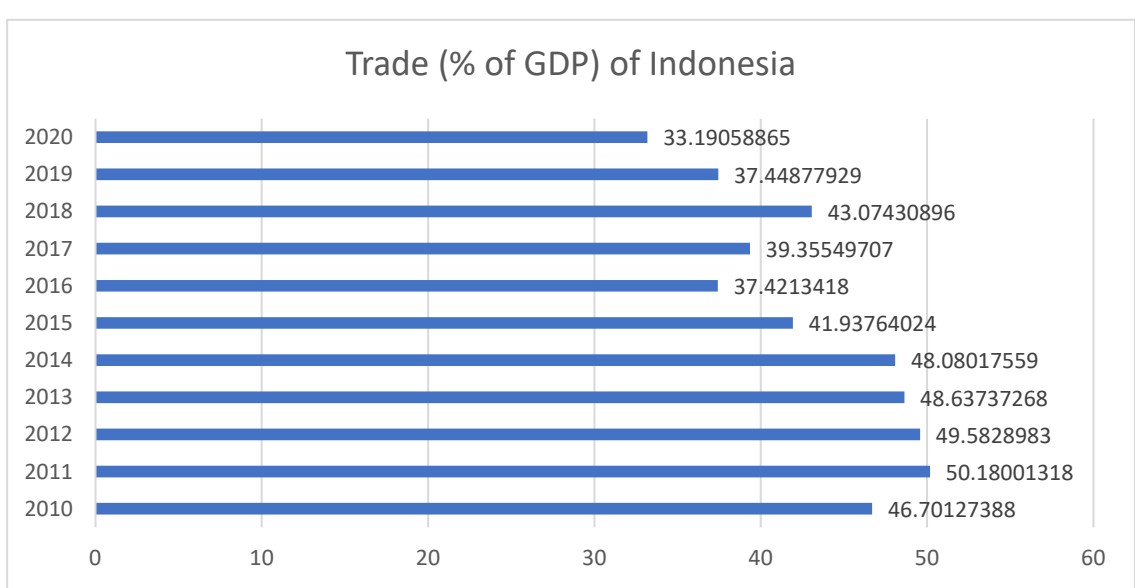

**Figure 3.** Trade openness of Indonesia (2010–2020). Source: authors modified the chart from the world bank data on World Bank Group, 2022. (https://data.worldbank.org/indicator/NE.TRD.GNFS.ZS?end=2020&locations=ID&start=2010, Indonesia's trade percentage of GDP, accessed on 12 February 2022).

*2.2. The Link between Trade and Food Security*

There are certain pathways through which households experience the effect of trade, with varying impacts on different dimensions of food security such as food availability, accessibility, and utilization. In order to simplify the complexity of the food security concept, we follow the concept developed in previous research [23] and focus only on the three dimensions of aggregate food availability, household food accessibility and individual-level food intake.

International trade can impact the accessibility of food through lower food prices and greater household incomes because international trade may enhance economic growth and stimulate household incomes. Thus, trade can create employment, resulting in an increase in individual earnings, thereby enabling them to buy more food, in terms of quantity and diversity. Further, food access emphasizes the adequacy of household incomes to obtain a variety of food for household consumption. Food intake determines the nutritional status of an individual, which is influenced by household income to afford an adequate supply of food [24].

Next, international trade can be measured in terms of trade liberalization and trade openness, e.g. tariff rate and trade openness index [25]. Both measurements reflect similar characteristics of trade; however, trade openness measures the size of an economy's tradeable sector, while tariff rates are used to measure policies that reduce trade openness. Trade protections can also occur as non-tariff barriers, which include protections such as quota restrictions and labeling, and while measuring trade liberalization requires consideration of both tariff rates and non-tariff barriers, the difficulty in collecting and recording data on non-tariff barriers creates challenges in quantifying non-tariff barriers [16].

**3. Material and Method**

*3.1. Model and Estimation Strategy*

Referring to the baseline model developed in the previous studies [2,14,20] we specified the empirical model to estimate the relation between trade liberalization and food

security. Thus, it is assumed that food security is a function of trade liberalization and other inputs such as social and economic factors.

Accordingly, we assumed that food security could be approximated by the following production function.

FS = f (TLIB, X )

FS = f (AGRI,MANUF,ECON,SOC)

FS denotes food security and TLIB denotes trade liberalization measurement, X denotes the other variables that may affect food security. Meanwhile, trade liberalization is proxy with de jure measurement which is import tariffs, split into two sectoral exposure such as AGRI for the agriculture sector and MANUF for the food manufacturing sector; FS is measured in food nutrition intake per capita calorie and protein; ECON is for the economic variables and SOC is for the social variables.

The study follows the research framework shown in Figure 4 below:

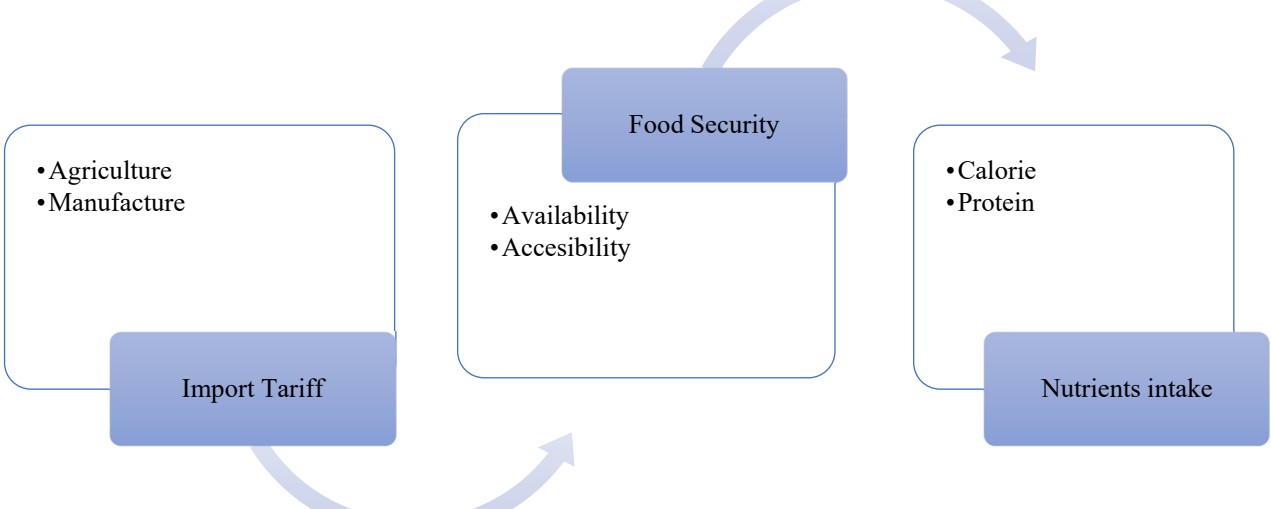

**Figure 4.** Research Framework. Source: authors.

We estimate the effect of trade on food security using energy intake and protein intake as a proxy for food security because food consumption is a key component in reducing food insecurity; it is determined by food availability and accessibility among the people in each district and is measured by daily per capita intake. Trade is measured by sectoral tariff exposure across districts in which the estimation is applied to all sample districts and differentiates between the districts in and outside Java and Bali.

Hence, we specified the model estimation to analyze the impact of trade liberalization on food security into several regression equations that are varied based on food security measurement such as calorie and protein per capita intake Further, based on the analysis of existing literature and to fulfill the purpose of this study, an empirical model using panel data fixed-effect regressions was constructed to determine the effect of trade liberalization on food security.

The model is as follows:

$$FS_{it} = \alpha + \beta_1 TRAGF_{it} + \varepsilon_{it} \tag{1}$$

$$FS_{it} = \eta + \tau_1 TRMF_{it} + \varepsilon_{it} \tag{2}$$

$$FS_{it} = \lambda + \rho_1 TRAGF_{it} + \rho_2 TRMF_{it} + \varepsilon_{it} \tag{3}$$

$$FS_{it} = \theta + \delta_1 TRAGF_{it} + \delta_2 TRMF_{it} + \sum_{k=3}^{K} \delta_k X_{3,it} + \varepsilon_{it} \tag{4}$$

Because the study aims to examine trade liberalization across Indonesia at the district level, the model examines each individual district i in time period $t$. FS denotes food security, which is measured by calorie and protein intake. Tariffs are differentiated between agriculture tariffs (TRAGF) and food manufacturing tariffs (TRMF). Tariff that is used in this estimation is simple average tariff, in which we choose the all-duty codes tariff. Index X refers to the set of control variables that may influence food security; the selected control variables include for instance GRDP per capita, agriculture population density, household average size, women's education. These can be categorized into two groups:

1. Economic variables: GRDP per capita, poverty, household employment share, income share.
2. Social variables: Agriculture population density, aggregate household size, education.

The estimation method is developed further by adding interaction variables (INT) in the control variables that interacting tariffs with the share of household employment in agriculture (SA) and in manufacturing (SM), and the share of household income based on their field of employment, where income from employment in agriculture is denoted ISA and income from manufacturing is denoted ISM. The developed model is as follows:

$$
\begin{aligned}
\text{FS}_{\text{it}} = \ \omega + \ & \vartheta_1 \text{TRAGF}_{\text{it}} + \vartheta_2 \text{TRMF}_{\text{it}} \\
& + \vartheta_3 INT\_SA + \vartheta_4 \text{INT\_SM} + \sum_{k=5}^{K} \vartheta_k X_{5,\text{it}} + \varepsilon_{\text{it}}
\end{aligned}
\tag{5}
$$

$$
\begin{aligned}
\text{FS}_{\text{it}} = \ \varphi + \ & \pi_1 \text{TRAGF}_{\text{it}} + \pi_2 \text{TRMF}_{\text{it}} \\
& + \pi_3 INT\_ISA + \pi_4 \text{INT\_ISM} + \sum_{k=5}^{K} \pi_k X_{5,\text{it}} + \varepsilon_{\text{it}}
\end{aligned}
\tag{6}
$$

Moreover, we also divide the sample into two main geographic areas based on population density: the islands of Java and Bali and the other islands in the archipelago (Sumatera, Kalimantan, Sulawesi Papua and other small island groups). Thus, the model in Equation (4) is estimated in which the districts are grouped based on the island groups. This estimation is performed because the effect of tariff might be varied across districts on different islands.

In consideration to non-tariff barriers as the new form of tariff measurement, which is not considered in this study due to a lot of time and effort is needed in gathering and measuring the data. The same is true for openness data, which is limited to national data only. Hence, this study measures trade liberalization only using import tariff rates, with sectoral import tariffs being measured based on their exposure to each district in Indonesia. We use all-duty codes tariff, it is not only for MFN (Most Favor Nation) tariff. The tariffs are categorized into two sectors: agriculture and manufacturing that related to food sector.

Following the method developed by Kis Katos and Sparrow [26], tariff exposure is measured based on economic sector composition and is defined by weighting tariff lines according to the sectoral shares in each district's regional GDP [26]. Using the Equation (7), we convert the national tariff of agriculture sector and food manufacturing sector to district level.

The equation is as follows:

$$
T_{k,t}^{GRDP} = \sum_{h=1}^{H} \left( \frac{GRDP_{h,k}}{GRDP_{h,2000}} \times T_{ht} \right)
\tag{7}
$$

where $T$ represents the tariff in each district $k$ and period $t$ and is generated from the national tariff $T$ per sector $h$ weighted by the sectoral GRDP (Gross Regional Domestic Product) in each district $k$. To construct $T$ (tariff exposure), the import tariff is weighed by the GRDP per district. We use the same procedure in calculating each tariff exposure of agriculture and food manufacturing sector. In this study, instead of using overall tariffs, we prefer to use the sectors that are related to food imports, i.e., the agriculture and food manufacturing sectors; hence, the overall tariff has been divided into two sectors. The separation of those

sectoral tariffs is because the impact raised by each sector might be different. The tariffs are weighted by gross regional domestic product to account for variations in the exposure effect across different districts. Accordingly, each region has different economic composition, which may result in different tariffs exposure which can be expressed in terms of total output [26].

*3.2. Data and Analysis*

　　This study uses 2011 to 2018 consumption and GRDP data from Central Bureau of Statistics (BPS). We collected data beginning from 2011 because the yearly basis consumption data start from that year, before which the survey data was available only once in three years. Furthermore, the period between 2011 to 2018 was marked by relatively low tariff rates, making it an ideal time to examine the effect of trade liberalization on food security. The import tariffs of each district were calculated and categorized into two sectors according to the equation in the previous subsection.

　　SUSENAS (Indonesian National Socio Economic Survey) data from BPS were used to measure food security through total calorie and protein intake per capita per day. The total number of districts for which the data are available is 496, which are later split into districts from Java and Bali island and other districts outside Java. The quality of food consumed by people determines the nutrient status in which household food consumption depends on the income of household that defines its purchasing power. There are many nutrients in the food consumed such as calories, protein, fat and carbohydrates; however, only calories and protein are used as a proxy of food security in this case. These two nutrients are important for individual dietary needs. Calories are the expression of energy measurement and protein is related to enzyme and body immune. BPS (Statistic Indonesia) used the level of calorie and protein consumed to express the level of nutritional adequacy which is one of the indicators of population welfare.

　　In order to estimate the relationship between trade and food security, district fixed effect regressions were performed for the time period from 2011 to 2018. As a proxy for trade liberalization, we used the tariff import data gathered from UNCTAD Trade Analysis Information System (TRAINS). We also used data on GRDP per capita, poverty rate, agriculture population density and household size as control variables. The explanatory variables and control variables were based on the developed model, the construction of which was guided by past studies on trade and food security. As the estimation involves time series panel data, the model was regressed using the fixed effect panel data regression method, and comparisons are drawn between the sample of all districts and that of districts in islands outside Java and Bali. Three models were estimated, with the following specifications:

　　The first model estimates solely the effect of tariffs on food intake in which the effect of each tariff is estimated in different equation.

$$\text{Cal}_{it} = \alpha^{calagr} + \beta_1^{calagr} \text{TARGF}_{it} + \varepsilon_{it} \tag{8}$$

$$\text{Cal}_{it} = \alpha^{calmanu} + \beta_1^{calmanu} \text{TRMF}_{it} + \varepsilon_{it} \tag{9}$$

$$\text{Prot}_{it} = \alpha^{protagr} + \beta_1^{protagr} \text{TRAGF}_{it} + \varepsilon_{it} \tag{10}$$

$$\text{Prot}_{it} = \alpha^{protmanu} + \beta_1^{protmanu} \text{TRMF}_{it} + \varepsilon_{it} \tag{11}$$

　　In the second model, the effect of both tariffs is included in the estimation equation.

$$\text{Cal}_{it} = \eta^{cal} + \tau_1^{cal} \text{TARGF}_{it} + \tau_2^{cal} \text{TRMF}_{it} + \varepsilon_{it} \tag{12}$$

$$\text{Prot}_{it} = \eta^{prot} + \tau_1^{prot} \text{TARGF}_{it} + \tau_2^{prot} \text{TRMF}_{it} + \varepsilon_{it} \tag{13}$$

The third model estimates the effect of tariffs on food intake with control variables:

$$
\begin{aligned}
\text{Cal}_{it} = \ & \lambda^{cal} + \rho_1^{cal}\text{TARGF}_{it} + \rho_2^{cal}\text{TRMF}_{it} + \rho_3^{cal}\text{l\_gcap}_{it} + \rho_4^{cal}\text{hh}_{\text{size}it} \\
& + \rho_5^{cal}\text{pov}_{it} + \rho_6^{cal}\text{l\_popfis}_{it} + \varepsilon_{it}
\end{aligned}
\tag{14}
$$

$$
\begin{aligned}
\text{Prot}_{it} = \ & \lambda^{prot} + \rho_1^{prot}\text{TARGF}_{it} + \rho_2^{prot}\text{TRMF}_{it} + \rho_3^{prot}\text{l\_gcap}_{it} + \\
& \rho_4^{prot}\text{hh\_size}_{it} + \rho_5^{prot}\text{pov}_{it} + \rho_6^{prot}\text{l\_popfis}_{it} + \varepsilon_{it}
\end{aligned}
\tag{15}
$$

The fourth model estimates the effect of tariffs on food intake with interaction variables for tariffs in which the model construction of which was guided by the past studies on trade and food security.

$$
\begin{aligned}
\text{Cal}_{it} = \ \theta^{cal} + \ & \delta_1^{cal}\text{TARGF}_{it} + \delta_2^{cal}\text{TRMF}_{it} + \delta_3^{cal}\text{l\_gcap}_{it} + \delta_4^{cal}\text{hh}_{\text{size}it} \\
& + \delta_5^{cal}\text{pov}_{it} + \delta_6^{cal}\text{l\_popfis}_{it} + \delta_7^{cal}\text{TRAGF}*\text{SA} + \delta_8^{cal}\text{TRMF} \\
& *\text{SM} + \varepsilon_{it}
\end{aligned}
\tag{16}
$$

$$
\begin{aligned}
\text{Prot}_{it} = \ \theta^{prot} + \ & \delta_1^{prot}\text{TARGF}_{it} + \delta_2^{prot}\text{TRMF}_{it} + \delta_3^{prot}\text{l\_gcap}_{it} + \delta_4^{prot}\text{hh}_{\text{size}it} \\
& + \delta_5^{prot}\text{pov}_{it} + \delta_6^{prot}\text{l\_popfis}_{it} + \delta_7^{prot}\text{TRAGF}*\text{SA} \\
& + \delta_8^{prot}\text{TRMF}*\text{SM} + \varepsilon_{it}
\end{aligned}
\tag{17}
$$

$$
\begin{aligned}
\text{Cal}_{it} = \ \gamma^{cal} + \ & \sigma_1^{cal}\text{TARGF}_{it} + \sigma_2^{cal}\text{TRMF}_{it} + \sigma_3^{cal}\text{l\_gcap}_{it} + \sigma_4^{cal}\text{hh}_{\text{size}it} \\
& + \sigma_5^{cal}\text{pov}_{it} + \sigma_6^{cal}\text{l\_popfis}_{it} + \sigma_7^{cal}\text{TRAGF}*\text{ISA} + \sigma_8^{cal}\text{TRMF} \\
& *\text{ISM} + \varepsilon_{it}
\end{aligned}
\tag{18}
$$

$$
\begin{aligned}
\text{Prot}_{it} = \ \gamma^{prot} + \ & \sigma_1^{prot}\text{TARGF}_{it} + \sigma_2^{prot}\text{TRMF}_{it} + \sigma_3^{prot}\text{l\_gcap}_{it} + \sigma_4^{prot}\text{hh}_{\text{size}it} \\
& + \sigma_5^{prot}\text{pov}_{it} + \sigma_6^{prot}\text{l\_popfis}_{it} + \sigma_7^{prot}\text{TRAGF}*\text{ISA} \\
& + \sigma_8^{prot}\text{TRMF}*\text{ISM} + \varepsilon_{it}
\end{aligned}
\tag{19}
$$

We use natural log for certain variables, such as agriculture population density and GRDP per capita, in order to obtain better estimates. Tariff data is measured by sector due to the limited availability of data on import tariff per food product, particularly at the district level.

Table 1 presents the list of variables and measurement units used in the models:

**Table 1.** List of Variables and Measurement.

| Variable | Description | Unit of Measurement |
|---|---|---|
| Tariff (Sectoral) | Import tariff<br><br>- Agricultural Tariff<br>- Food Manufacturing Tariff | Percentage (All duty codes) |
| Food Security | - Calorie Consumption<br>- Protein Consumption | Average kilocalorie/capita/day<br>Average gram protein/capita/day |
| Control Variables | GRDP per capita, Agriculture population density, Household average size, education, poverty rate | In natural log of GDP, natural log of agriculture population, share of the wife's education in the family (level junior high) |
| Interaction Variables 1<br>Interaction Variables 2 | Share of people working in agriculture; Share of people working in manufacturing<br>Share of household income in Agriculture and Manufacturing field | In percentage based on SUSENAS data |

## 4. Results and Discussion

### 4.1. Descriptive Statistics

The following Figure 5 plots the national import tariff rates from 2011 to 2018 for the two sectors that are related to food imports.

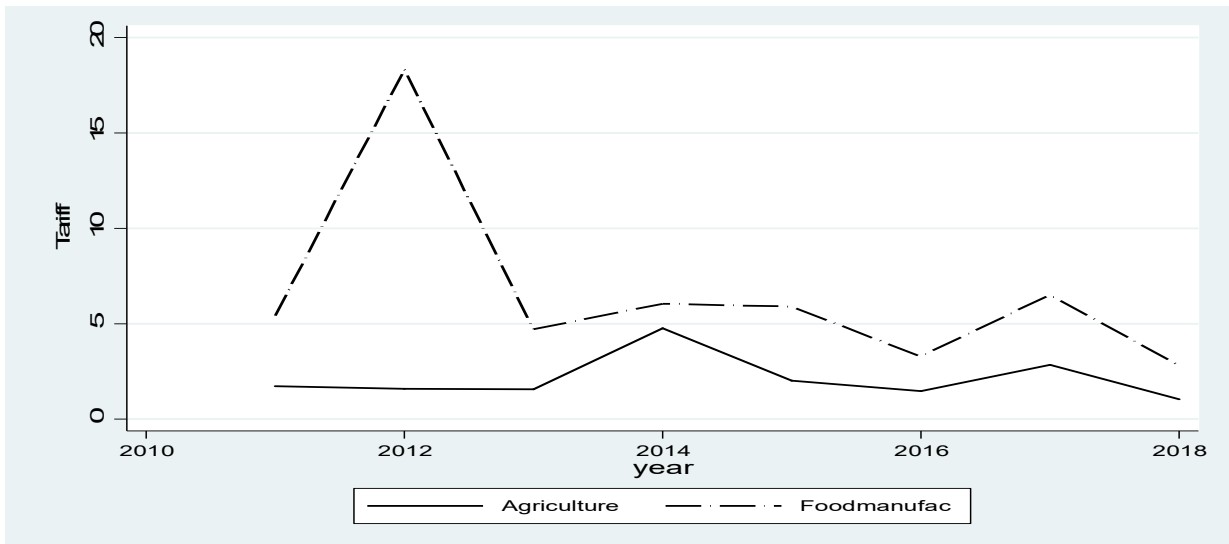

**Figure 5.** National sectoral tariff for agriculture and manufacturing sectors.

Despite some spikes and slight fluctuations in Indonesia's agriculture and manufacturing sector tariffs, both sectors exhibit a trend of declining tariffs at the national level. Yet, the graph in Figure 6 illustrates that overall tariff exposure at the regional level increased moderately between 2011 and 2018 for both sectors. Agriculture tariff exposure declined slightly after 2011 but fluctuated upwards between 2014 to 2018, whereas manufacturing tariff exposure saw greater fluctuations throughout the entire period of 2011 to 2018. This depicts the mean of both tariff exposure of regions per year, which in general fluctuate over the years.

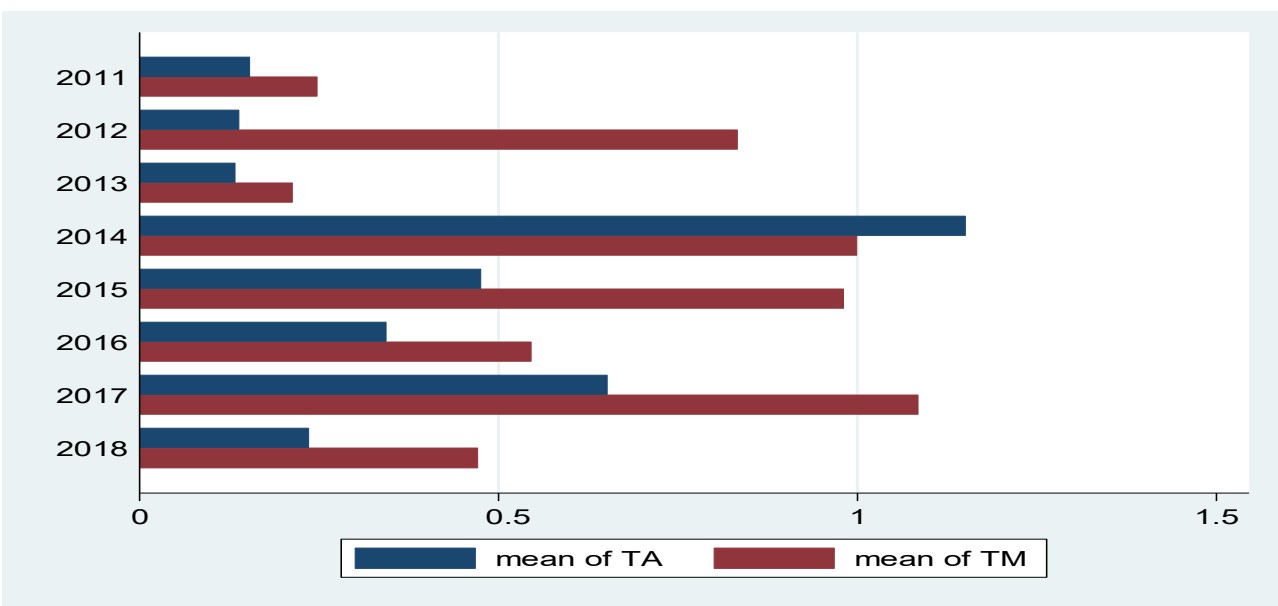

**Figure 6.** Mean of regional tariff exposure per year for the agriculture (TA) and food manufacturing (TM) sectors.

Next, the following map in Figure 7 presents the average consumption of calories (AkE-the threshold of calorie consumption) and proteins (AkP-the threshold of protein consumption) across districts in Indonesia in 2018. The map distinguishes regions according to whether average consumption in the region surpasses the level of sufficient nutrition intake as stipulated by the Ministry of Health: 2150 kcal for calories and 57 grams of protein.

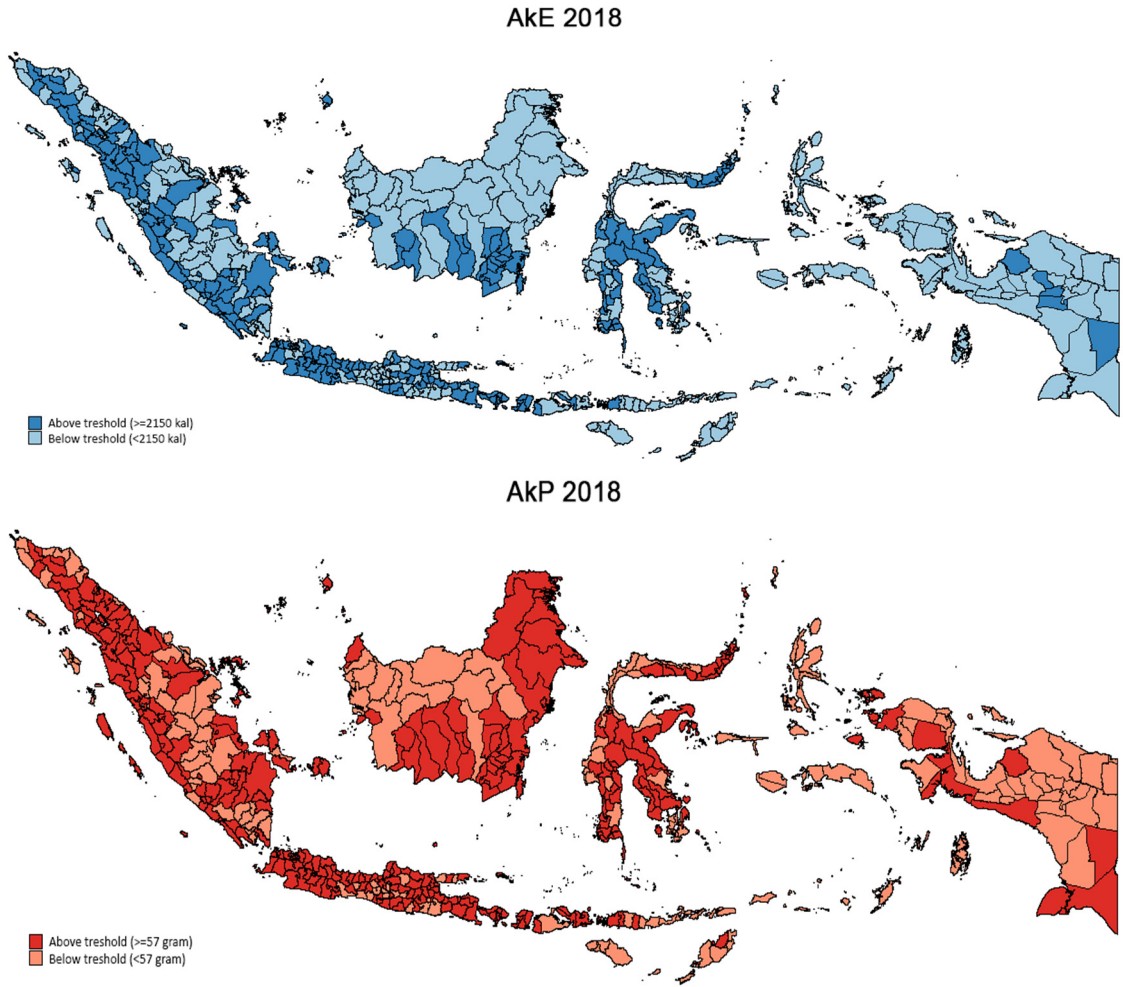

**Figure 7.** Map of Calorie and Protein threshold intake 2018.

The map (see Figure 7) reveals that most districts in Eastern Indonesia have not satisfied the minimum threshold of calorie and protein consumption. In contrast, most districts in Java and Bali have achieved the minimum threshold. The stark differences in nutritional intake between the two regions are among the reasons why we also provide separate estimates for the relationship between tariff exposure and food security in the sample that excludes Java and Bali.

In addition, the graph in Figure 8 illustrates the range of calorie and protein consumption between 2011 and 2018 for districts in each province. As mentioned, there are some regions that have not met the sufficient level of dietary need; however, the range of nutrition intake between those time period is slightly declining across provinces. This result is depicted in the graph in Figure 8.

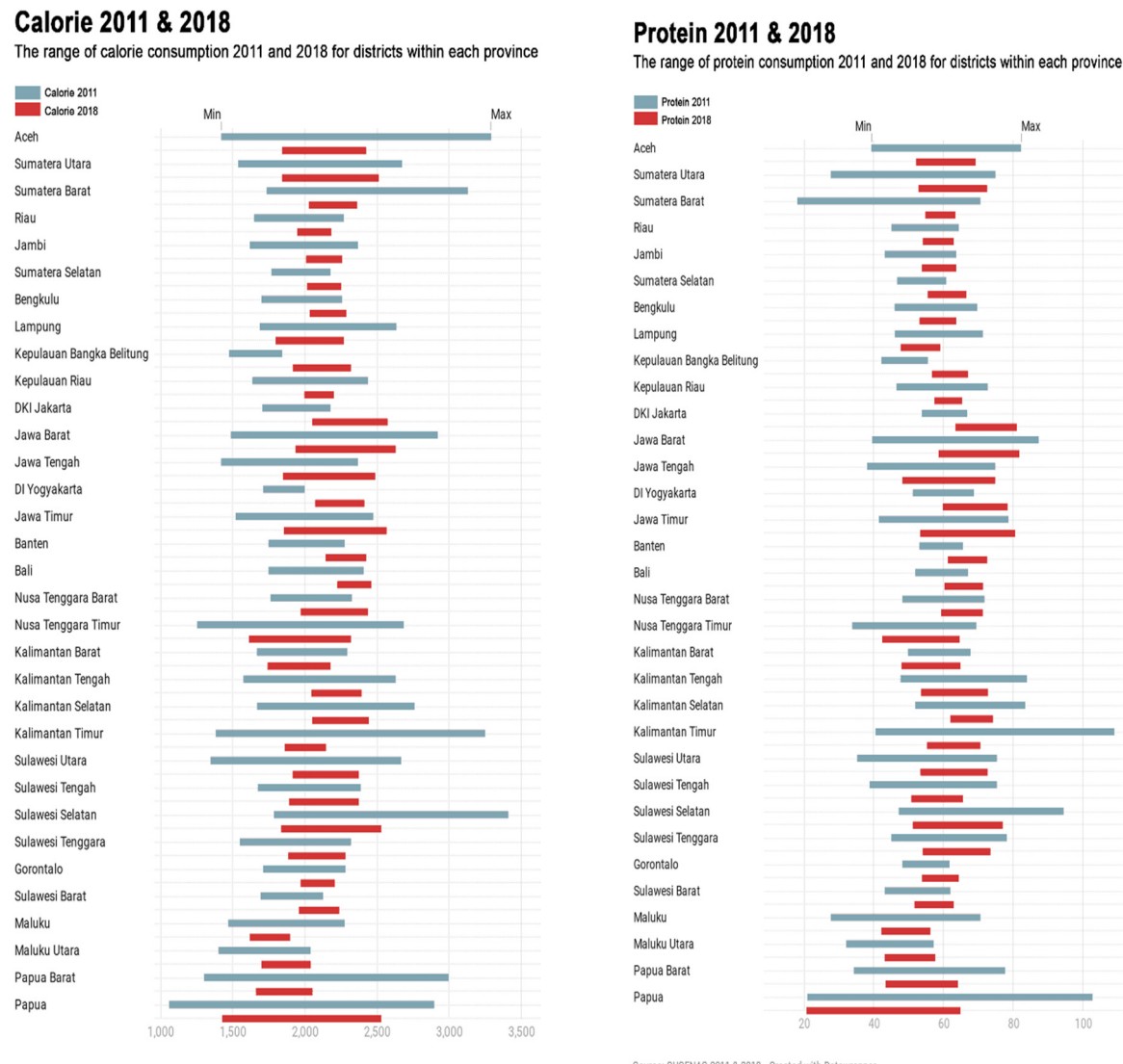

**Figure 8.** The range of calorie and protein consumption between 2011 and 2018 for districts within provinces.

It can be seen from the graphs (Figure 8) that most provinces saw a decrease in their calorie and protein consumption gap between 2011 to 2018. This is following the trend of national import tariffs that are also going down, as shown in Figure 5. Nonetheless, the relation between tariff and food security cannot be concluded solely from these graphs as there are many other factors that may have contributed to food security.

*4.2. Estimation Results*

The results in both Tables 2 and 3 show that tariff protection may negatively affect food security; an increase in both agriculture and manufacturing tariffs resulted in a decrease of average household nutrient intake, i.e., a decrease in calorie (see Table 2) and protein (see Table 3) consumption per capita. The interaction factors refer to the share of those who work in the agriculture and manufacturing sectors and the income share of household work in agriculture and manufacturing. Tariffs are estimated separately in the first and second column, while in the third to last column, both tariffs are included.

As can be seen from Table 2, the results in the first and second columns indicate that agriculture and manufacturing tariffs have negative and significant coefficients towards nutrient intake. The coefficients are consistently negative and significant even when both

tariffs are considered jointly in the third column and when more control variables and interaction terms are included in the regression. The result indicates that an increase in tariff exposure is associated with a decrease in calorie consumption, but the size of the effect is different between agriculture and manufacturing sector tariffs.

The impact of manufacturing tariff exposure is stronger than that of agriculture tariff exposure: the results in column 3 imply that the increase of agriculture and manufacturing tariffs by a one-unit percentage will reduce calorie consumption by 31.26 kcal and 38 kcal, respectively. The coefficient of agriculture is less than that of manufacturing tariffs. There are slightly different results when interaction variables are included; agriculture tariff exposure has a larger impact on calorie consumption.

**Table 2.** Estimation results calorie model.

| | Dependent Variable: Calorie | | | | | |
|---|---|---|---|---|---|---|
| | **Model 1** | | **Model 2** | **Model 3** | **Model 4** | |
| Independent Variables | (1) Agriculture Tariff | (2) Manufacture Tariff | (3) Both Tariffs | (4) No Interaction Variables | (5) Interaction Variables 1 Employment Share | (6) Interaction Variables 2 Income Share |
| TRAGF | −31.26 *** (4.839) | | −31.12 *** (4.749) | −19.23 *** (4.688) | −65.88 *** (13.88) | −58.56 *** (12.77) |
| TRMF | | −38.00 *** (3.346) | −37.94 *** (3.325) | −28.39 *** (3.296) | −42.01 *** (5.589) | −41.76 *** (5.615) |
| lgcap | | | | 70.87 *** (11.97) | 77.08 *** (12.07) | 76.01 *** (12.05) |
| hh_size | | | | −220.1 *** (30.66) | −210.9 *** (30.63) | −210.4 *** (30.65) |
| povertyrate | | | | −15.70 *** (3.175) | −15.56 *** (3.165) | −15.40 *** (3.167) |
| lpfis1 | | | | 34.26 *** (8.290) | 33.87 *** (8.270) | 33.90 *** (8.277) |
| Women Educ (SMP) | | | | 290.7 *** (81.49) | 305.2 *** (81.30) | 302.9 *** (81.32) |
| TRAGF * AgricultureShare | | | | | 181.6 *** (51.13) | |
| TRMF * ManufactureShare | | | | | 186.1 ** (60.88) | |
| TRAGF * AgriIncome | | | | | | 170.6 ** (51.91) |
| TRMF * ManufacIncome | | | | | | 179.3 ** (59.94) |
| _cons | 2005.6 *** (5.869) | 2006.0 *** (4.203) | 2038.0 *** (6.421) | 2682.5 *** (153.9) | 2628.2 *** (153.9) | 2628.3 *** (154.0) |
| $N$ | 3851 | 3850 | 3850 | 3569 | 3569 | 3569 |
| $R^2$ | 0.012 | 0.037 | 0.049 | 0.123 | 0.129 | 0.129 |

Standard errors in parentheses * $p < 0.05$, ** $p < 0.01$, *** $p < 0.001$.

**Table 3.** Estimation result for protein model.

| | Dependent Variable: Protein | | | | | |
|---|---|---|---|---|---|---|
| | Model 1 | Model 2 | Model 3 | Model 4 | | |
| Independent Variables | (1) Agriculture Tariff | (2) Manufacture Tariff | (3) Both Tariffs | (4) No Interaction Variables | (5) Interaction Variables 1 Employment Share | (6) Interaction Variables 2 Income Share |
| TRAGF | −0.729 *** (0.153) | | −0.726 *** (0.151) | −0.508 *** (0.151) | −1.554 *** (0.449) | −1.338 ** (0.413) |
| TRMF | | −0.892 *** (0.106) | −0.891 *** (0.106) | −0.694 *** (0.106) | −0.990 *** (0.181) | −0.978 *** (0.182) |
| lgcap | | | | 1.298 *** (0.386) | 1.437 *** (0.390) | 1.406 *** (0.390) |
| hhsize | | | | −4.124 *** (0.989) | −3.922 *** (0.990) | −3.919 *** (0.991) |
| povertyrate | | | | −0.585 *** (0.102) | −0.582 *** (0.102) | −0.578 *** (0.102) |
| lpfis | | | | 1.351 *** (0.267) | 1.343 *** (0.267) | 1.343 *** (0.268) |
| Women Educ (SMP) | | | | 17.96 *** (2.629) | 18.28 *** (2.628) | 18.22 *** (2.629) |
| TRAGF * AgricultureShare | | | | | 4.073 * (1.653) | |
| TRMF * ManufactureShare | | | | | 4.047 * (1.968) | |
| TRAGF * AgriIncome | | | | | | 3.605 * (1.678) |
| TRMF * ManufacIncome | | | | | | 3.819 * (1.937) |
| _cons | 55.92 *** (0.186) | 55.94 *** (0.134) | 56.69 *** (0.205) | 67.91 *** (4.966) | 66.71 *** (4.975) | 66.76 *** (4.978) |
| N | 3851 | 3850 | 3850 | 3569 | 3569 | 3569 |
| $R^2$ | 0.007 | 0.021 | 0.027 | 0.087 | 0.090 | 0.090 |

Standard errors in parentheses * $p < 0.05$, ** $p < 0.01$, *** $p < 0.001$.

The tariff has a greater effect on consumption for households working either in the agriculture or manufacturing sector. The minimizing tariff effect on consumption is slightly larger for those who work in agriculture.

Table 3 explores the effect of tariffs on protein consumption. The baseline model is presented in column 4, while columns 1 and 2 show the individual effect of each tariff and column 3 shows their effects when they are included jointly. Columns 5 and 6 show results that include interaction variables between tariff exposure and employment and income shares. The results show that agriculture and manufacturing tariffs have negative impacts on protein consumption, with the baseline model predicting that protein consumption declines by 0.5 grams and 0.7 grams when tariffs for agriculture and manufacturing sectors are increased by one percentage point respectively.

These results are consistent with the tariff effects on calorie consumption, where the impact of manufacturing tariffs is greater than that of agriculture tariffs. Meanwhile, the regressions in columns 5 and 6 show that the impact of the tariff is larger when there is an interaction factor of people working in those sectors. The effect of the interaction factor towards consumption may minimize the negative effect of tariff raises, although the coefficient of interaction becomes less significant, falling to the 5 percent level.

Overall, the results in Tables 2 and 3 show that the effect of tariffs on calorie and protein consumption is negative, which implies that when tariffs increase, consumption decreases. We also find that when tariffs interact with the share of employment and the share of income, the effect of tariffs on calorie and protein intake is higher than the result without interaction variables. It also indicates that the effect of interaction variables itself have minimized the decline of consumption. As for the control variables, the effect on food consumption corroborates the theoretical framework and results from previous studies.

Next, referring to the control variables included in the regressions, Figure 4 has demonstrated that geographical aspects such as location and the remoteness of the area may also affect the resource endowments of each district and therefore contribute to the level of food security in the area [27]. Hence, to explore those variations and to ensure that the results from the overall sample are not driven only by the socioeconomic characteristics of the large metropolitan areas in Java and Bali, we divide the sample into five subgroups based on the major island groups in Indonesia: (1) Sumatera; (2) Jawa and Bali; (3) Kalimantan; (4) Sulawesi; and (5) Nusa Tenggara, Maluku and Papua, which we take as the reference group. The estimation result is presented in Table 4.

**Table 4.** Estimation based on sample composition of five islands.

| Sub Group | Sumatera | | Jawa/Bali | | Kalimantan | | Sulawesi | | NTB/NTT/Maluku/ Papua | |
|---|---|---|---|---|---|---|---|---|---|---|
| | (1) | | (2) | | (3) | | (4) | | (5) | |
| | Calorie | Protein | Calorie | Protein | Calorie | Protein | Calorie | Protein | Calorie | Protein |
| TRAGF | −7.759 | −0.364 | −2.384 | 0.378 | −45.56 * | −1.447 * | −8.274 | −0.163 | −67.00 *** | −1.515 *** |
| | (7.002) | (0.220) | (10.99) | (0.377) | (22.08) | (0.714) | (10.45) | (0.331) | (14.23) | (0.457) |
| TRMF | −29.59 *** | −0.635 ** | −20.99 *** | −0.527 *** | −38.69 *** | −1.076 ** | −46.30 ** | −1.064 * | −23.20 | −0.219 |
| | (6.705) | (0.211) | (3.648) | (0.125) | (10.49) | (0.339) | (17.02) | (0.539) | (20.14) | (0.647) |
| lgcap | 95.23 *** | 1.794 * | 103.3 *** | 2.587 *** | −31.87 | −3.285 ** | 86.86 ** | 1.992 * | 7.780 | 0.427 |
| | (23.11) | (0.727) | (21.60) | (0.740) | (39.11) | (1.265) | (29.88) | (0.945) | (32.80) | (1.054) |
| hhsize | −235.7 *** | −4.005 * | −387.9 *** | −10.20 *** | −254.8 * | −5.899 | −289.0 *** | −5.278 *** | 3.665 | 1.914 |
| | (55.78) | (1.755) | (61.08) | (2.093) | (101.7) | (3.288) | (78.29) | (2.477) | (73.40) | (2.359) |
| povertyrate | −14.83 * | −0.764 *** | −27.67 *** | −0.994 *** | −20.06 | −1.356 * | 13.30 | 0.303 | −13.29 * | −0.244 |
| | (5.921) | (0.186) | (5.681) | (0.195) | (18.84) | (0.609) | (13.23) | (0.419) | (6.063) | (0.195) |
| lpfis | 34.45 * | 0.878 * | 4.355 | −0.189 | 19.48 | 1.077 | 45.91 | 1.816 | 32.53 | 1.895 ** |
| | (14.09) | (0.443) | (23.03) | (0.790) | (20.36) | (0.658) | (36.39) | (1.151) | (17.79) | (0.572) |
| Women Educ | 65.84 | 11.69 ** | 429.1 ** | 18.90 *** | 142.9 | 17.19 * | 289.5 | 17.39 * | 578.7 * | 26.38 *** |
| | (137.8) | (4.335) | (139.1) | (4.768) | (255.0) | (8.246) | (234.2) | (7.410) | (232.4) | (7.470) |
| _cons | 2712.8 *** | 70.14 *** | 3324.5 *** | 95.71 *** | 3212.2 *** | 97.24 *** | 2603.9 *** | 60.27 *** | 1965.6 *** | 35.59 ** |
| | (291.7) | (9.175) | (292.2) | (10.01) | (459.1) | (14.84) | (426.3) | (13.49) | (402.3) | (12.93) |
| N | 1148 | 1148 | 968 | 968 | 409 | 409 | 560 | 560 | 484 | 484 |
| $R^2$ | 0.137 | 0.090 | 0.287 | 0.219 | 0.085 | 0.086 | 0.095 | 0.052 | 0.090 | 0.091 |

Standard errors in parentheses* $p < 0.05$, ** $p < 0.01$, *** $p < 0.001$.

The sign of the tariff coefficients remains the same for all groups, albeit with a markedly greater effect for the NTT/NTB/Maluku and Papua islands and with different significance across other islands. Agriculture tariffs are only significant in Kalimantan and Nusa Tenggara/Maluku and Papua, whereas the manufacturing tariff coefficient is significant in all islands except Nusa Tenggara/Maluku and Papua. Thus, tariff effects in and Nusa Tenggara/Maluku and Papua are driven by agriculture tariffs, whereas manufacturing tariffs drive most of the effects in Sumatera, Java/Bali and Sulawesi. These differences might be due to a variety of local conditions and cultures in each district, which can induce different responses to tariff changes.

Thus, this study confirmed some findings from previous studies [14,16]; for example, it is found that trade openness and dietary energy consumption are positively associated, and the latter concludes that lower trade barriers positively impact calorie and protein intake. Furthermore, this study also finds that trade liberalization can have negative effects on calorie and protein intake, with varying effects across different districts and islands.

Some control variables in this study are also worth noting. For example, the education level of women in the family positively influences food consumption, as women often play a larger role in providing the food and dietary needs of households. It is also found

that employment in agriculture and manufacture is helpful in minimizing the negative influence of rising tariffs, especially as the agriculture sector is still a key employment sector in Indonesia, with 31.86% of Indonesians in 2017 being employed in the sector and therefore relying on agriculture for their livelihoods [28]

Thus, if tariffs in both sectors rise, calorie and protein consumption may decrease and adversely affect the welfare of households. This study found evidence that the coefficients for both tariffs are negative and significant, indicating that higher sectoral tariffs are associated with lower food consumption. These effects vary across islands; in remote islands such as Nusa Tenggara, Maluku and Papua, agriculture tariffs have a greater impact than manufacturing tariffs, and the reverse is true in more populated and urban islands such as Java, Bali and Sumatera.

## 5. Conclusions

This study examined the relationship between food security (which is defined as calorie and protein intake) and trade liberalization (which is defined as the tariffs in the agriculture and food manufacturing sectors). This study found that calorie and protein consumption as a measurement of food security is affected by tariff protection. Import tariffs in food are negatively associated with calorie and protein consumption, but the effect of trade liberalization—i.e., the effect of agriculture and manufacture sectoral tariff exposure—on food consumption may be minimized if households work in the agriculture and/or manufacturing sectors. It can also be said that tariffs, particularly agriculture tariffs, are likely to have a greater impact on households living in remote or rural areas—areas in which poor households are often concentrated. Thus, trade liberalization may exacerbate the vulnerability that poor households have towards food insecurity, in which from the data Indonesia's food security index is increasing nowadays.

We find that the impact of tariffs on food consumption is lower in Java's districts compared to those outside Java. It is also found that food consumption in the districts in Sumatera, Bali/Jawa and Sulawesi are driven more by food manufacturing tariffs, while those in the other islands are affected by agriculture tariffs. Therefore, decisions with regard to tariff hikes should ideally consider the subsequent effect on people and varying effects of tariff exposure that may widen the inequalities between urban and rural districts. From a policy perspective, this calls for greater consideration of geographic locations, local economic conditions, and the share of employment in each district in order to design better tariff policies.

This study differs from previous studies in two ways. First, we derived tariff exposure into two sectors that are related to food consumption. Second, we identified the association between tariffs and food security by looking into subgroups based on geographic locations. Our main results confirm that tariffs on agriculture and food manufacturing negatively impact food security. We also interact tariff variables with the employment and income share in each sector to show that the negative effects of tariffs toward calorie and protein intake might be outweighed by the positive effects from the employment share of each sector, which are rooted in the income that households earn from working in those sectors.

This study has several limitations and can be further developed to consider tariffs for certain food products that are consumed by a majority of households in Indonesia and to consider other forms of tariff and non-tariff measurements. Another limitation is that consumption data are not available on a yearly basis before 2011, thereby restricting the years available for our analysis. Further insight may also be garnered by estimating the effect of trade openness on food security if export–import data at the district level are available. This might be a more sensible measurement of trade; however, it is also challenging to gather the data at the district level. Nonetheless, this study has contributed to the debate regarding the relationship between import tariffs and food insecurity at the sub-national level, particularly in Indonesia, and this study has highlighted the need for governments to consider more closely the protection of vulnerable households from the effect of tariffs and the possibility of using tariffs as a channel to reduce food insecurity. By supporting

investment in domestic agriculture and food manufacturing sectors, governments may reduce the harmful effects of tariffs through the boosting of household incomes, especially for households in remote areas.

**Author Contributions:** Conceptualization, M.H.M., M.E., T.D., D.W., M.P.; methodology, M.H.M., M.E., T.D., D.W., M.P.; software, M.H.M.; validation, M.E., T.D., D.W.; formal analysis, M.H.M., T.D., D.W.; investigation, M.H.M., D.W.; resources and data, M.H.M.; writing-original draft preparation, M.H.M., T.D.; writing-review and editing, M.H.M., M.E.; visualization, M.H.M., M.E., T.D., D.W.; supervision, T.D., M.E., D.W. All authors have read and agreed to the published version of the manuscript.

**Funding:** This research received no external funding.

**Institutional Review Board Statement:** Not applicable.

**Informed Consent Statement:** Not applicable.

**Data Availability Statement:** Not applicable.

**Acknowledgments:** The authors thank and extend their appreciation to Faculty of Economics University of Indonesia for the contribution in providing the data and giving access to references used in writing the paper.

**Conflicts of Interest:** The authors declare no conflict of interest.

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
