# Peer review of "The Analysis of Trade Liberalization and Nutrition Intake for Improving Food Security across Districts in Indonesia"

_sustainability, doi:10.3390/su14063291_

Round 1
Reviewer 1 Report
1-Please make the same format in the abstract section as like other sections.
2- its better to add some future direction in your abstarct section.
3- Need to add reference in the first paragraph of introduction.
4- Please arrange your article reference style according to the journal format.
5- Its a research article not a thesis. please summarize and remove review of literature.
6- Please pay more attention on your M&M section.
7- Its better to revise the figure. 3.
8- Overall, article need extensive revision for possible publication.
9- please read some recent articles and follow the MDPI formatting style for your article.
Author Response
February 26th , 2022
Dear Reviewer 1,
Thank you for submitting a revised article entitled Import Tariff and Nutrition Intake: Study across districts in Indonesia.
Following the reviewer's recommendation, I changed the title of the article to The Analysis of Trade Liberalization and Nutrition Intake for Improving Food Security Study across districts in Indonesia.
Here I present the revision to the reviewer 1. There are several changes according to the revision list provided. Update on the article using track changes.
Thank you very much for your consideration.
Yours Sincerely,
Mahjus Ekananda
University of Indonesia
Kampus UI Depok 16424, Indonesia. www.ui.ac.id
Tel.: +62-21-78849152; Fax: +62-21-78849154
E-mail: mahyusekananda@gmail.com
FIRST REVIEWER
Open Review
(x) I would not like to sign my review report
( ) I would like to sign my review report
English language and style
( ) Extensive editing of English language and style required
( ) Moderate English changes required
(x) English language and style are fine/minor spell check required
( ) I don't feel qualified to judge about the English language and style
|
Yes |
Can be improved |
Must be improved |
Not applicable |
|
|
Is the content succinctly described and contextualized with respect to previous and present theoretical background and empirical research (if applicable) on the topic? |
( ) |
(x) |
( ) |
( ) |
|
Are the research design, questions, hypotheses and methods clearly stated? |
( ) |
(x) |
( ) |
( ) |
|
Are the arguments and discussion of findings coherent, balanced and compelling? |
( ) |
(x) |
( ) |
( ) |
|
For empirical research, are the results clearly presented? |
( ) |
(x) |
( ) |
( ) |
|
Is the article adequately referenced? |
( ) |
(x) |
( ) |
( ) |
|
Are the conclusions thoroughly supported by the results presented in the article or referenced in secondary literature? |
( ) |
(x) |
( ) |
( ) |
|
|
|
|
|
|
Attachment :
Reviewer’s Comments and Suggestion and Authors’ Responds
Point 1.
Please make the same format in the abstract section as like other sections.
Response 1.
We have changed the format of abstract to times new roman 12pt.
Point 2.
It’s better to add some future direction in your abstract section.
Response 2.
Future direction added to the last line of abstract. From line 31 to 33.
Point 3.
Need to add reference in the first paragraph of introduction.
Response 3.
We add reference that support the statement in the first paragraph
Point 4.
Please arrange your article reference style according to the journal format.
Response 4.
We have rectified the reference following the journal format. We follow Vancouver citation followed Sustainability format
Point 5.
It’s a research article not a thesis. please summarize and remove review of literature.
Response 5.
The subtitle is removed and there are some parts that we make it simple.
Point 6.
Please pay more attention on your M&M section.
Response 6.
We made some changes in this section such as the equation typo has been edited. We also added explanation for each equation model.
Point 7.
It’s better to revise the figure. 3.
Response 7.
We revised the number of the figure and corrected all the figure that is not in the right places and add to the manuscript.
Point 8.
Overall, article need extensive revision for possible publication.
Response 8.
There are changes made in the introduction, the descriptive result, the result table, the model equations, add citation, the theoretical background, and the references.
Point 9.
Please read some recent articles and follow the MDPI formatting style for your article.
Response 9.
We have referred to certain article published in journal of MDPI (i.e. Mary, 2018; Fusco et al, 2020, Azizah et al, 2022) and changed the article style such as the sections’ title.

Reviewer 2 Report
1-
Definitions of the attached graphics can be given. X-Y axes are not open. In addition, since the data are given in the final form of 2020 and we are now in 2022, it would be good if the last information until the last year could be given.
2-
I saw grammatical errors in English in the summary. You can use active sentences (we did, researched, etc.) when writing your scientific article summary if necessary, but keep in mind that most of your abstract may require passive sentence structures (done, researched, etc.).
3-
I saw errors in expression and tense in the introductory sentence.
While global population growth and food growth are almost equal, the disproportion between regions with high food production and population distribution causes an imbalance in access to nutrition. (Line 35-37)
4-
Line 45 I couldn't find any reference to who is Ricardo. It would be nice to have a source.
5-
First of all, the definition should be made.
Food security is the need for people to have access to food. Food security is a measure of the availability of food and individuals' accessibility to it, where accessibility includes affordability. There is evidence of food security being a concern over 10,000 years ago, with central authorities in ancient China and ancient Egypt being known to release food from storage in times of famine. At the 1974 World Food Conference the term "food security" was defined with an emphasis on supply. Food security, they said, is the "availability at all times of adequate, nourishing, diverse, balanced and moderate world food supplies of basic foodstuffs to sustain a steady expansion of food consumption and to offset fluctuations in production and prices.
6-
I would like to see data and information from this report in your article. This is a good resource.
https://docs.wfp.org/api/documents/WFP-0000130141/download/?_ga=2.147367991.1623111103.1645308380-1964226351.1645308380
7-
It should also be in the summary given at the end, and there should be detailed information about the subject in the summary.
8-
The title should also be able to express the whole subject better. The title can also be deduced from the result.
9-
The time used in the conclusion must have passed. Because the work has been done.
10-
How reliable can the data of who consumes what be? (Line 237) Health data could also be added to the model. If the subject of accessing and purchasing food is examined here, people can also be fed with cheaper protein-free and calorie-free foods. (Line 241)
11-
Do the indicators called agriculture and food cover processed and unprocessed foods?
12-
Imported food product chart is given, but it is unclear from which country which food goes to which region. I would love to see this. In addition, protein and calories are mentioned, but there is no meat and milk in the table.
13-
If 5 regions of Indonesia were examined in the study, this could be written in the title. Likewise, it would be nice if this information was given in the summary, conclusion and model design sections.
14-
New publications on the subject can be added. The Indonesian economy can be given in more detail.
https://webcache.googleusercontent.com/search?q=cache:oFuR31_xa-8J:https://repository.cips-indonesia.org/media/341329-negative-effects-of-non-tariff-trade-bar-010dd7e2.pdf+&cd=12&hl=tr&ct=clnk&gl=tr
Author Response
February 26th , 2022
Dear Reviewer 2,
Thank you for submitting a revised article entitled Import Tariff and Nutrition Intake: Study across districts in Indonesia.
Following the reviewer's recommendation, I changed the title of the article to The Analysis of Trade Liberalization and Nutrition Intake for Improving Food Security Study across districts in Indonesia.
Here I present the revision to the reviewer 2. There are several changes according to the revision list provided. Update on the article using track changes.
Thank you very much for your consideration.
Yours Sincerely,
Mahjus Ekananda
University of Indonesia
Kampus UI Depok 16424, Indonesia. www.ui.ac.id
Tel.: +62-21-78849152; Fax: +62-21-78849154
E-mail: mahyusekananda@gmail.com
SECOND REVIEWER
Open Review
(x) I would not like to sign my review report
( ) I would like to sign my review report
English language and style
( ) Extensive editing of English language and style required
( ) Moderate English changes required
( ) English language and style are fine/minor spell check required
(x) I don't feel qualified to judge about the English language and style
|
Yes |
Can be improved |
Must be improved |
Not applicable |
|
|
Is the content succinctly described and contextualized with respect to previous and present theoretical background and empirical research (if applicable) on the topic? |
( ) |
(x) |
( ) |
( ) |
|
Are the research design, questions, hypotheses and methods clearly stated? |
( ) |
(x) |
( ) |
( ) |
|
Are the arguments and discussion of findings coherent, balanced and compelling? |
( ) |
(x) |
( ) |
( ) |
|
For empirical research, are the results clearly presented? |
( ) |
(x) |
( ) |
( ) |
|
Is the article adequately referenced? |
( ) |
(x) |
( ) |
( ) |
|
Are the conclusions thoroughly supported by the results presented in the article or referenced in secondary literature? |
( ) |
(x) |
( ) |
( ) |
Attachment :
Reviewer’s Comments and Suggestion and Authors’ Responds
Point 1
Definitions of the attached graphics can be given. X-Y axes are not open. In addition, since the data are given in the final form of 2020 and we are now in 2022, it would be good if the last information until the last year could be given.
Response 1.
The chart in figure 1 has been corrected by given the label for X and Y axes. Unfortunately, the latest GSFI data has not available yet. There are some changes in the chart and graph attached, the order of the figures is changes.
Point 2
I saw grammatical errors in English in the summary. You can use active sentences (we did, researched, etc.) when writing your scientific article summary, if necessary, but keep in mind that most of your abstract may require passive sentence structures (done, researched, etc.).
Response 2.
We have made some changes to the sentence’s structures in abstract to passive sentence based on the suggestion (please find in the line 22-33).
Point 3a
I saw errors in expression and tense in the introductory sentence.
Response 3a.
We have changed the sentence to correct the grammar error. (please find it in 37-39).
Point 3b
While global population growth and food growth are almost equal, the disproportion between regions with high food production and population distribution causes an imbalance in access to nutrition. (Line 35-37)
Response 3b.
We changed the grammar in the line mentioned which can be found in line 37-39. We follow the changes that suggested here. (Thank you)
Point 4
Line 45 I couldn't find any reference to who is Ricardo. It would be nice to have a source.
Response 4.
It is David Ricardo’s theory of comparative advantage. We have changed the sentence and put the citation. (line 53-55)
Point 5
First of all, the definition should be made.
Food security is the need for people to have access to food. Food security is a measure of the availability of food and individuals' accessibility to it, where accessibility includes affordability. There is evidence of food security being a concern over 10,000 years ago, with central authorities in ancient China and ancient Egypt being known to release food from storage in times of famine. At the 1974 World Food Conference the term "food security" was defined with an emphasis on supply. Food security, they said, is the "availability at all times of adequate, nourishing, diverse, balanced and moderate world food supplies of basic foodstuffs to sustain a steady expansion of food consumption and to offset fluctuations in production and prices.
Response 5.
The definition is added to second paragraph, and we also added citation for the definition which we put it in footnote. (Thank you)
Point 6
I would like to see data and information from this report in your article. This is a good resource.
Point 6a.
https://docs.wfp.org/api/documents/WFP-0000130141/download/?_ga=2.147367991.1623111103.1645308380-1964226351.1645308380
Response 6a.
We have add the data of PoU as figure 1 in line 133.
Point 6
Response 6b.
The definition of food security is cited from this source.
Point 7
It should also be in the summary given at the end, and there should be detailed information about the subject in the summary.
Response 3.
It is added in the conclusion.
Point 8
The title should also be able to express the whole subject better. The title can also be deduced from the result.
Response 3.
The title is changed based on the suggestion given. The title is reflected more the subject in the research.
Point 9
The time used in the conclusion must have passed. Because the work has been done.
Response 9.
It is changed to past tense, and we deleted the unnecessary sentences.
Point 10
How reliable can the data of who consumes what be? (Line 237) Health data could also be added to the model. If the subject of accessing and purchasing food is examined here, people can also be fed with cheaper protein-free and calorie-free foods. (Line 241)
Response 10.
It is based on the data SUSENAS which is the individual consumption per capita and from those data we aggregate to per district.
About the health data we have used it in the other paper that link to malnutrition because the data availability is not the same with the data set in this paper.
Point 11
Do the indicators called agriculture and food cover processed and unprocessed foods?
Response 11.
The indicator of agriculture is for the import that fall into the category of agriculture sector which seclude forestry sub sector. Food is for manufacturing in food sector covered all food import.
Point 12
Imported food product chart is given, but it is unclear from which country which food goes to which region. I would love to see this. In addition, protein and calories are mentioned, but there is no meat and milk in the table.
Response 12.
The chart is only taken from statistic Indonesia that show the overall value of total import of Indonesia of certain food production. The data of the country where the food is imported, unfortunately is not available in this research. In Indonesia the import product goes through the main port which is not all regions has it. From the main port then the product is distributed to each region. Meanwhile the data about the value of import from one port to the other port within regions is not available hence this becomes one of the limitations of this paper.
Point 13
If 5 regions of Indonesia were examined in the study, this could be written in the title. Likewise, it would be nice if this information was given in the summary, conclusion, and model design sections.
Response 13.
The model is the same as equation 4 that is estimated based on the group of islands and mentioned in the design sections. The information is written in the conclusion and also mentioned the result in the abstract.
Point 14a
New publications on the subject can be added. The Indonesian economy can be given in more detail.
Response 14a.
We have read the article and it has a lot of information needed. We have cited this working paper into this paper (line 140-143).
Point 14b
https://webcache.googleusercontent.com/search?q=cache:oFuR31_xa-8J:https://repository.cips-indonesia.org/media/341329-negative-effects-of-non-tariff-trade-bar-010dd7e2.pdf+&cd=12&hl=tr&ct=clnk&gl=tr
Response 14b.
It is cited in our paper. (Thank you)

Round 2
Reviewer 1 Report
can be considered for publication.
Author Response
March 3rd , 2022
Dear Reviewer 1,
Thank you for submitting a revised article entitled The Analysis of Trade Liberalization and Nutrition Intake for Improving Food Security Study across districts in Indonesia.
Here I present the revision to the reviewer 1. There are several changes according to the revision list provided. Update on the article using track changes.
Thank you very much for your consideration.
Yours Sincerely,
Mahjus Ekananda
University of Indonesia
Kampus UI Depok 16424, Indonesia. www.ui.ac.id
Tel.: +62-21-78849152; Fax: +62-21-78849154
E-mail: mahyusekananda@gmail.com
FIRST REVIEWER
Open Review
(x) I would not like to sign my review report
( ) I would like to sign my review report
English language and style
( ) Extensive editing of English language and style required
( ) Moderate English changes required
(x) English language and style are fine/minor spell check required
( ) I don't feel qualified to judge about the English language and style
Point 1: English language and style are fine/minor spell check required
Response 1: We have made some changes in the grammar. We checkhed the grammar error using the Gramarly application.
Editor Comments: Citation of references: both in the bibliography and in the text of the manuscript (especially in the case of figures) should be prepared according to the Journal editorial requirements.
Response to editor : We have made changes to the reference format and the figures.
